# OLD N-GRAMS NEVER DIE: TOWARDS IDENTIFYING LLMS-GENERATED TEXT USING ANTIQUE N-GRAMS

## ABSTRACT

The proliferation of large language models (LLMs) has triggered an influx of AI-generated content, making robust detection of such content paramount for maintaining academic, journalistic, and regulatory integrity. However, the community has largely overlooked a time-tested resource that classical n-gram models, trained exclusively on human-authored corpora, may serve as a de facto gold standard for identifying machine-generated writing. In this paper, we build upon well-trained pre-AI N-Gram models to form the backbone of a lightweight AI-text detection system called **GramGuard**. Specifically, by generating paraphrased variants via temperature-controlled decoding from LLMs, we measure the shifts in log-likelihood, entropy, and token frequency variance between original texts and perturbed versions. These *delta* features then feed into an ensemble classifier to yield interpretable decisions about authorship. Extensive experiments on PubMed, WritingPrompts, and XSum demonstrate that **GramGuard** matches or exceeds state-of-the-art detectors in performance and robustness. Our findings reaffirm the enduring value of pre-AI n-gram models and introduce a scalable, transparent solution for AI-text detection. The code and datasets are released at
`https://github.com/N-Gram-dev/GramGuard`.

## 1 INTRODUCTION

The rapid advancement of large language models (LLMs) has ushered in a new era of AI-generated text that increasingly saturates digital communication spaces Brown et al. (2020). With capabilities that rival or even surpass those of human experts in fluency and coherence, LLMs are now widely employed to produce persuasive news articles, academic essays, and algorithmically generated contentWu et al. (2025). While these models hold tremendous potential, they also introduce profound social risks, such as news fabrication and academic ghostwritten submissions Kumarage et al. (2024). These societal implications have motivated significant efforts in AI text detection, leading to a growing body of research aimed at distinguishing machine-generated content from human-written text Zellers et al. (2019); Chakraborty et al. (2024). As LLMs evolve, their outputs become increasingly indistinguishable from natural human texts, narrowing the detectable gaps that early detectors once relied upon Fang et al. (2025); Krishna et al. (2023). Despite continual methodological improvements, existing detection approaches often struggle to keep pace with the sophistication of modern generative models, highlighting a pressing need for more robust and resilient solutions Zhou et al. (2025).

The state-of-the-art detection techniques encompass a variety of approaches, particularly the mainstreaming of curvature statistics-driven zero-shot classifiers Mitchell et al. (2023); Bao et al. (2024); Ma & Wang (2024). For instance, a recent study attempted to reconstruct truncated text and then inspect the N-Gram-wise difference between the original text and its reconstructed versions in a black-box manner Yang et al. (2024). While these methods have demonstrated promising gains in detecting AI text, they are also sensitive to the architectural idiosyncrasies of specific LLMs. In contrast, the enduring legacy of pre-AI n-gram language models that were merely trained on human-authored text has been largely overlooked Brants et al. (2007); Heafield (2011). These "antique" models encode pure human linguistic patterns, rendering them inherently adversarial to machine-generated sequences. Upon that, we raise an inspiration: **could such "antique" n-gram models exclusively trained on human text serve as a natural *"gold standard"* for machine-generated**

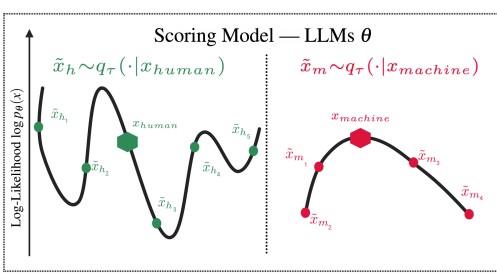 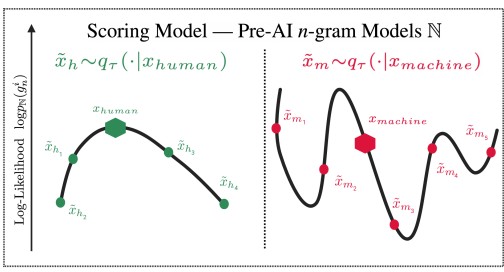

Figure 1: **LLM scoring vs. Pre-AI Ngram model scoring.** The log-likelihood tendencies of LLM-paraphrased human text $\tilde{x}_h \sim q_\tau(\cdot \mid x_{human})$ and Machine generated text $\tilde{x}_m \sim q_\tau(\cdot \mid x_{machine})$ compared with their original texts while using either LLMs (**left**) or Pre-AI Ngram model (**right**) for text probability scoring. $\tau$ is a source model used for text perturbation.

**text detection?** The intrinsic strength of N-gram models lies in establishing a statistically rigorous baseline for human-written distributions, which offer inherent sensitivity to statistical deviations in AI-generated texts, thereby enabling discrimination through probabilistic divergences, entropy anomalies, and other measures Shannon (1948); Jurafsky & Martin (2023). Furthermore, their computational efficiency, low resource requirements, and interpretable decision metrics further establish them as practical, transparent tools for lightweight detection frameworks.

Contemporary detection paradigms claim that large language models tend to preferentially generate tokens with elevated conditional likelihoods due to their probability-biased sampling mechanisms Gehrmann et al. (2019). This proclivity yields sequences that exhibit low perplexity and increased curvature in the log-probability landscape Bao et al. (2024); Fang et al. (2025). In contrast, human authors compose text with intent, rather than maximizing probability, which likely results in more diffuse token-wise distributions. Under perturbation, the process of rephrasing machine text tends to sample the tokens with lower probabilities compared with their original sample. Yet, such a phenomenon is uncertain in human text. Building on this consensus, we have the following corollary:

**Corollary 1.1.** *When adopting an n-gram model trained exclusively on human corpora for probability scoring, human-written text tends to exhibit higher n-gram log-likelihoods than machine-generated content. Upon perturbation, the log-likelihood of human text typically decreases consistently, reflecting disruption of human-style linguistic patterns. In contrast, rewritten machine text often yields smaller or inconsistent changes in n-gram probability, indicating statistical rigidity or instability under perturbation.*

Fig. 1 demonstrates the N-Gram-wise log-likelihood discrepancies between human texts and AI ones after perturbation, which is opposite to the phenomenon of mainstream LLM-based scoring models. Specifically, when scoring using a pre-AI n-gram model (see right side of Fig. 1) exclusively trained on human corpora, human text exhibits higher log-likelihoods and undergoes a consistent drop in likelihood across variants. Yet, AI text shows lower likelihoods and erratic shifts upon paraphrasing. To verify the feasibility of the above assertion, we present a novel N-Gram-based AI text detection framework – **GramGuard** to identify whether a text is generated from a specific model. Specifically, we measure how the N-Gram properties of a text shift from an N-Gram perspective after LLM perturbation via: (1) text paraphrasing using LLM under various decoding temperature settings; (2) Text N-Gram-wise scoring using N-Gram models trained on human corpora with a backoff strategy, where the probability of an N-Gram is the log-likelihood of its last token by taking the N-1 prefix into account; (3) discrepancy analysis based on three interpretable metrics: *log-likelihood shifts*, *entropy changes*, and *token frequency variance deltas*; (4) decision-making by feeding these features into a shallow classifier for prediction. The main contributions of this study are threefold:

- We propose using pre-LLM KenLM n-gram models as a "*gold standard*" for detecting AI-generated text, given their exclusive exposure to human-authored corpora.
- We design GramGuard, a delta-based framework that computes N-Gram-wise log-likelihood, entropy, and frequency variance shifts across paraphrased variants to reveal stylistic rigidity.
- Our method achieves state-of-the-art detection accuracy across three datasets and maintains robustness under paraphrastic attacks, while requiring only CPU-based inference.

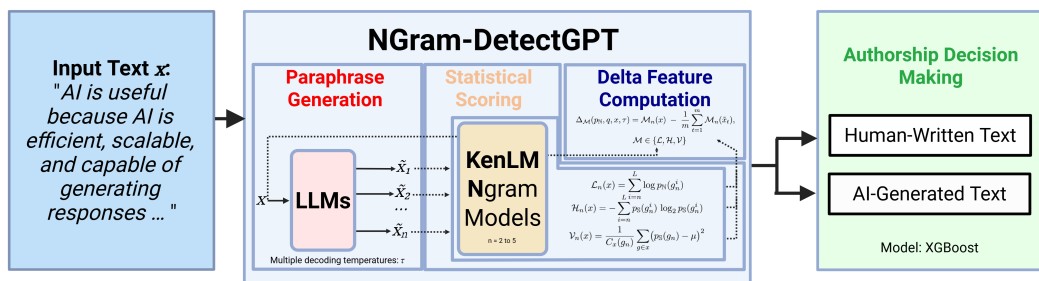

Figure 2: Overview of the **GramGuard** detection pipeline. Input text is paraphrased using LLMs across multiple decoding temperatures, evaluated by fixed KenLM n-gram models, and scored across three statistical metrics: log-likelihood, entropy, and frequency variance. Delta-based deviations are computed between the original and variants, and these features are classified using a lightweight ensemble XGBoost for AI authorship prediction.

## 2 RELATED WORK

The rapid proliferation of large language models (LLMs) has prompted an urgent need for robust detection systems capable of distinguishing between human-authored and AI-generated content Kumarage et al. (2024). Research in this area spans four methodological paradigms: watermarking-based detection, probability-based scoring, statistical feature analysis, and hybrid ensemble approaches Wu et al. (2025). Watermarking methods embed imperceptible signals during the text generation process. Early examples, such as logit-based token biasing Kirschenbauer et al. (2024), were followed by improvements like context-aware token partitioning Guo et al. (2024b) and SBERT-based rejection sampling in SEMSTAMP Hou et al. (2024). Despite their innovation, watermarking methods degrade under paraphrasing and domain shifts, as demonstrated by recent work on watermark collisions Luo et al. (2025) and adversarial stress tests Zhou et al. (2025).

Likelihood-based detectors exploit model curvature patterns Mitchell et al. (2023), with Fast-DetectGPT Bao et al. (2024) offering faster inference through curvature binarization. These methods, however, suffer from temperature sensitivity and model mismatch, and are vulnerable to paraphrastic attacks Krishna et al. (2023), Cheng et al. (2025). In contrast, statistical approaches use token distributions, entropy, and rank features to detect text anomalies Yang et al. (2024), Wu et al. (2023), Gehrmann et al. (2019). These build on classical stylometry methods Cavnar & Trenkle (1994), Burrows (2002), Stamatatos (2006), which analyzed authorship using n-gram frequencies and function-word variance. More recent hybrid detectors such as StackMore Gritsai et al. (2024) and contrastive paraphrase filters Fang et al. (2025) Guo et al. (2024a) attempt to fuse these paradigms, but still face reproducibility and robustness challenges. In this context, our delta-based approach offers an interpretable and lightweight alternative that leverages paraphrastic shifts in fluency, entropy, and frequency variance, yielding competitive performance under black-box conditions.

## 3 TASK SETTINGS

### 3.1 DETECTION ASSUMPTIONS AND PROBLEM SETUP

We propose **GramGuard**, a hybrid detection framework for binary classification. Unlike state-of-the-art methods such as DetectGPT and Fast-DetectGPT that rely on zero-shot and white-box assumptions Zhu et al. (2023), our approach operates in a supervised black-box setting. We measure how N-Gram properties of text, specifically log-likelihoods, entropy, and frequency variance shifts under paraphrastic transformations induced by powerful LLMs.

We adopt KenLM-based n-gram models, trained exclusively on pre-LLM human corpora, as a gold standard language model, denoted as $p$. Given an input text $x$, we compute the delta between the original and its paraphrased variants $\tilde{x} \sim q_\tau(\cdot \mid x)$, where $q_\tau$ represents an LLM decoder at temperature $\tau$. We aim to capture systematic rigidity or instability in scoring behavior, enabling reliable authorship classification through shallow classifiers.

A key hypothesis behind **GramGuard** is that the human-written text in the current age aligns more closely with older human corpora than machine-generated text does. This reflects findings from Bao et al. (2024), where LLMs are shown to prefer tokens with higher model probabilities. We formalize this stylistic deviation assumption as:

$$
\overbrace{\boldsymbol{\Delta}(x_h, p_{\mathbb{N}}, q) \; = \; \log p_{\mathbb{N}}(x_h) \; - \; \mathbb{E}_{\tilde{x}_h \sim q_\tau(\cdot | x_h)} \log p_{\mathbb{N}}(\tilde{x}_h)}^{\textit{Variation of human text after paraphrasing }_{(\Delta_h)}}
$$

$$>$$

$$
\underbrace{\boldsymbol{\Delta}(x_m, p_{\mathbb{N}}, q) \; = \; \log p_{\mathbb{N}}(x_m) \; - \; \mathbb{E}_{\tilde{x}_m \sim q_\tau(\cdot | x_m)} \log p_{\mathbb{N}}(\tilde{x}_m)}_{\textit{Variation of machine text after paraphrasing}_{(\Delta_m)}}
$$

$$(1)$$

Here, $x_h$ and $x_m$ denote human and machine-generated texts, respectively. The scoring model $p_{\mathbb{N}}$ is a classical n-gram language model trained on pre-LLM human corpora, and $q_\tau$ represents the LLM-based paraphrasing function under decoding temperature $\tau$. This equation expresses the core assumption that human-authored text exhibits greater stylistic and lexical shift under paraphrasing, reflected as a larger drop in n-gram log-likelihood compared to machine-generated text.

### 3.2 Pre-AI N-Gram Preparation

To ensure stylistic purity and reliable scoring, we construct four KenLM-based n-gram models trained exclusively on human-authored corpora predating LLMs. Specifically, we use official 3-gram and 4-gram models from the *LibriSpeech* Language Modeling benchmark Panayotov et al. (2015), and train additional 2-gram and 5-gram models using KenLM's `lmplz` utility on the same corpus. All models are built on the `librispeech-lm-norm.txt.gz` dataset, sourced from the OpenSLR repository and comprising over 800 million words from 14,500 books in the Project Gutenberg archive. It is noteworthy that these texts, authored well before the rise of generative LLMs, offer a clean representation of human-authored language patterns. This uncontaminated foundation is essential to our detection framework, which compares statistical behaviors of input sentences before and after paraphrasing.

We then compute log-likelihood, entropy, and frequency variance scores across these models using N-Gram-embraced KenLM Heafield (2011), which are used to quantify deviations in linguistic structure and token distribution. This pretraining step ensures that all subsequent scoring reflects genuine human stylistic baselines, enhancing the sensitivity and interpretability of our delta-based metrics. In the next Section, we illustrate the steps of **GramGuard** that include: (1) text perturbation using LLM prompting under various decoding temperature settings; (2) N-Gram-based scoring using pre-AI N-Gram models via KenLM; (3) discrepancy quantification in perspectives of log-likelihood shifts, entropy changes, and token frequency variance deltas; (4) decision-making by feeding the three features into shallow classifiers for binary prediction.

## 4 GramGuard

To simulate realistic textual transformations and expose brittleness in synthetic content, we paraphrase each text by adopting the state-of-the-art LLM (*GPT-4_1-mini* OpenAI (2023) as the source model. The perturbed variants of data are acquired under a controllable decoding temperature: $\tau \in \{0.1, 0.3, 0.5, 0.7, 0.9, 1.1\}$. The variant sampling process can be formalized as:

$$\tilde{x}_t \sim q_\tau(\cdot \mid x), t \in [1, m] \tag{2}$$

where $q_\tau$ denotes a temperature-specified source morel, and $\tilde{x}_t$ is the $t$th variant of original text $x$ sampled via source model paraphrasing. The prompting templates for LLM perturbation are demonstrated in Appendix A (See the Supplementary Materials). Next, the N-Gram-wise statistical scoring will be accomplished via three features for "delta" analysis.

### 4.1 Text Scoring using N-Grams

Once paraphrased variants are generated, each text, either original or rewritten, is passed through four pre-trained KenLM n-gram models (2-gram to 5-gram) to extract three core statistical metrics:

*log-likelihood*, *Shannon entropy*, and *token frequency variance*. These metrics collectively characterize how fluency, lexical diversity, and repetition patterns shift under controlled perturbations.

**Log-Likelihood Estimation:** To quantify sequence-level fluency, we compute the conditional log-likelihood of each message using KenLM-based $n$-gram models. Let a tokenized message be defined as $x = \{w_1, w_2, \ldots, w_L\}$, where $w_i$ denotes the $i^{\text{th}}$ token and $L$ is the total number of tokens. Given an $n$-gram model $\mathcal{M}_n$, we define the $n$-gram ending at position $i$ as $g_n^i = \{w_{i-n+1}, \ldots, w_i\}$. The joint log-likelihood of the sequence $x$ is then:

$$\mathcal{L}_n(x) = \sum_{i=n}^{L} \log p_{\mathbb{N}}(g_n^i) \tag{3}$$

This formulation aggregates the log-probabilities of all overlapping $n$-grams in the sentence, providing a compact estimate of the sequence's fluency under an $n$-gram assumption. Each token $w_i$ is conditioned only on its $n-1$ predecessors, reflecting a Markovian assumption. For $n$-grams $g_n^i \notin \mathcal{M}_n$ (i.e., those with zero count in the training corpus), KenLM applies recursive backoff:

$$\log p_{\mathbb{N}}(g_n^i) = \begin{cases} \log p_{\mathbb{N}}(w_i | w_{i-n+1:i-1}), & \text{if } g_n^i \in \mathcal{M}_n \\ \lambda \cdot \log p_{\mathbb{N}}(g_{n-1}^i), & \text{otherwise} \end{cases} \tag{4}$$

Here, the probability of $g_n^i$ is calculated as the N-Gram conditional probability of $g_n^i$'s last token, $\alpha(g_{n-1}^i)$ is the backoff weight for its $(n-1)$-gram prefix, and $\lambda$ is a hyperparameter and Brants et al. (2007) suggests that $\lambda$ works well with the value of 0.4. Such a backoff sampling strategy allows an unknown N-Gram to access its lower-order Grams for conditional log probability estimation. KenLM precomputes these probabilities along with their backoff weights and stores them in ARPA-format binary tries for efficient lookup at inference time Heafield (2011). This $n$-gram log-likelihood scoring framework offers interpretable insights into sequence-level fluency by modeling how predictable a token sequence is with respect to human-authored corpora that predate modern LLMs.

**Statistical Entropy:** While log-likelihood captures sequence-level fluency, it does not fully characterize the stylistic footprint of a sentence. To provide a more nuanced view of n-gram structure, we additionally extract two statistics from texts' empirical n-gram distribution: *Shannon Entropy* Venkatraman et al. (2024), which quantifies lexical diversity and unpredictability, and *NGram frequency variance*, which reflects the unevenness or burstiness of NGram repetition. Let $G_{\mathbb{N}}$ denote the *N-Gram Vocabulary* of the dataset $\mathcal{D}$ that is used for training the scoring model - $p_{\mathbb{N}}$. We define the statistical probability of an n-gram $g_n^i \in G_{\mathbb{N}}$ as:

$$p_{\mathbb{S}}(g_n^i) = \frac{C_{\mathcal{D}}(g_n^i)}{\sum_{g_n \in G_{\mathbb{N}}} C_{\mathcal{D}}(g_n)} \tag{5}$$

where $C_{\mathcal{D}}(g_n^i)$ is the raw frequency count of $g_n^i$ in $\mathcal{D}$. In this case, $p_{\mathbb{S}}(g_n^i)$ is the estimated N-Gram frequency probability based on the total number of N-Grams in the scoring model. Thus, the Shannon Entropy of a given text is calculated as:

$$\mathcal{H}_n(x) = -\sum_{i=n}^{L} p_{\mathbb{S}}(g_n^i) \cdot \log_2 p_{\mathbb{S}}(g_n^i) \tag{6}$$

where $\mathcal{H}_n(x)$ captures the spread or uncertainty across the $x$'s n-gram usage. Higher entropy suggests richer and more diverse lexical usage, while lower entropy reflects repetitiveness and rigid phrasing often characteristic of machine-generated outputs under low sampling temperatures.

**N-Gram Frequency Variance:** In addition to Entropy, we estimate the discrepancy between text and perturbed versions in terms of N-gram frequency variance to offer a complementary perspective on lexical distribution. In other words, entropy reflects the overall diversity of N-gram usage, while variance highlights irregularities in frequency, such as repeated or dominant patterns. Together, these two measures provide a more complete view of structural consistency within text. To complement

this, we also compute N-Gram frequency variance, which captures the dispersion of individual n-gram occurrences. Let the mean n-gram frequency be:

$$\mu_n(x) = \frac{1}{L - n + 1} \cdot \sum_{i=n}^{L} p_{\mathbb{S}}(g_n^i) \tag{7}$$

where, $L$ represents the number of tokens in $x$, and the index $i$ denote the last token of a N-Gram in $x$. That means, the number of N-Grams in $x$ equals $L - n + 1$. Then, the N-Gram frequency variance of a given text is calculated as:

$$\mathcal{V}_n(x) = \frac{1}{C_x(g_n)} \cdot \sum_{g \in x} (p_{\mathbb{S}}(g_n) - \mu)^2 \tag{8}$$

Based on the previous process of scoring (1) N-Gram log likelihood $\mathcal{L}_n(x)$; (2) statistical entropy - $\mathcal{H}_n(x)$; and (3) N-Gram frequency - $\mathcal{V}_n(x)$; we formalize the discrepancies between the original text and its perturbed variants as follows.

$$\Delta_{\mathcal{M}}(p_{\mathbb{N}}, q, x, \tau) \leftarrow \mathcal{M}_n(x) - \frac{1}{m} \sum_{t=1}^{m} \mathcal{M}_n(\tilde{x}_t), \quad \mathcal{M} \in \{\mathcal{L}, \mathcal{H}, \mathcal{V}\} \tag{9}$$

where $p_{\mathbb{N}}$ is the KenLM-based N-Gram scoring model, $q$ is the source model for text perturbation, and $tau$ is the specific temperature worked on the source model $q$. Since $\mathcal{M} \in \{\mathcal{L}, \mathcal{H}, \mathcal{V}\}$, $\Delta_{\mathcal{L}}$ denotes the log-likelihood delta between the original text $x$ and its perturbed variants $\{\tilde{x}_1, ..., \tilde{x}_m\}$ that are sampled from a given source model $q$ with a specific sampling temperature $\tau$, vice versa for $\Delta_{\mathcal{H}}$ and $\Delta_{\mathcal{V}}$.

## 4.2 CLASSIFICATION

Following the extraction of delta-based features from paraphrased variants, we proceed to the final phase that aims to discriminate machine texts from human ones. The input (i.e.,$x$) to this stage is formalized as a twelve-dimensional vector, comprising twelve delta scores from log-likelihoods $\Delta_{\mathcal{L}}$, entropy $\Delta_{\mathcal{H}}$, and NGram frequency variance $\Delta_{\mathcal{V}}$ at four n-gram levels - $\{2, 3, 4, 5\}$. These features are carefully designed to capture statistical rigidity or flexibility under $\tau$-specified perturbation, and then fed into an interpretable ensemble-based classifier:**XGBoost (XGB)** Chen & Guestrin (2016), which is well-suited for tabular data and provides robustness to noise, built-in regularization, and feature importance attribution. It is noteworthy that XGBoost sequentially builds additive decision trees using second-order gradient descent and regularization to improve generalization and combat overfitting. In addition, the hyperparameters such as tree depth, learning rate, and number of estimators are tuned via grid search with 5-fold stratified cross-validation.

To ensure transparency and interpretability, we analyze feature importances derived from both models. These importances provide empirical insight into which delta metrics among fluency, entropy, and burstiness most strongly differentiate human writing from synthetic outputs. Classification performance is reported using ROC-AUC coupled with F1-score. This final step transforms nuanced statistical perturbation signals into reliable authorship attribution, offering a lightweight, interpretable, and robust solution for AI-generated text detection.

## 5 EXPERIMENTS

### 5.1 EXPERIMENTAL SETUPS

**Datasets:** We evaluate our framework on three established datasets spanning diverse domains Cornelius et al. (2024) and generation styles to ensure comparability with prior detection benchmarks Bao et al. (2024); Mitchell et al. (2023); Krishna et al. (2023). **PubMedQA** Jin et al. (2019), **Writing-Prompts** Fan et al. (2018), and **XSum** Narayan et al. (2018) include only human-authored samples. We use the public **TOCSIN** API-based release *as is* with its official splits.[1]

---

[1] `https://github.com/Shixuan-Ma/TOCSIN/tree/main/exp_API-based_model/data`

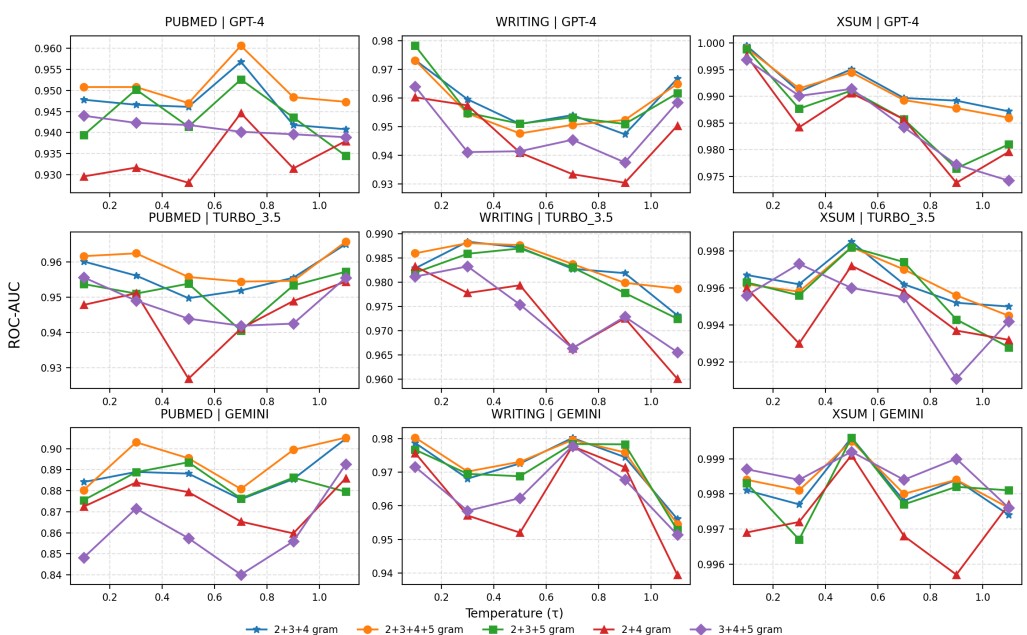

Figure 3: Temperature-wise ROC–AUC result curves across all nine datasets using various N-Gram combinations. It is noteworthy that the diagram only demonstrates the top-5 N-gram ensembles by mean AUC across the increased sampling temperatures from 0.1 to 1.1.

**Implementation details:** We synthesize AI responses using *GPT-4_1-mini* in a black-box manner. Texts are segmented at the sentence level for consistent scoring granularity. To simulate realistic perturbations, each sentence is paraphrased by all models under six decoding temperatures $\tau \in \{0.1, 0.3, 0.5, 0.7, 0.9, 1.1\}$, producing 10 variants per temperature. This controlled diversity enables robust delta-based analysis without requiring access to generation-time logits. For the scoring model, we adopt KenLM-based N-Gram models trained exclusively on human-authored corpora predating LLMs: the official LibriSpeech 3-gram and 4-gram models Panayotov et al. (2015), and custom 2-gram and 5-gram models built using `lmplz` Project (2015). These serve as stylistic baselines anchored in classical human language.

**Baselines:** We benchmark our detection approach against a range of leading detectors, including supervised classifiers (RoBERTa-Base, RoBERTa-Large), zero-shot scoring approaches including **DetectGPT** Mitchell et al. (2023), **Fast-DetectGPT** Bao et al. (2024), **LogRank** Krishna et al. (2023), **LRR** (an amalgamation of log probability and **Log-Rank**, other AI-generated-text detectors **DNA-GPT** Yang et al. (2024) and **GPTZero** Tian & Cui (2023), and statistical baselines using **entropy** and **likelihood** scores. We additionally include an internally implemented baseline - **NPR (Normalized Perplexity Rank)** for comparison completeness, which is derived from GLTR Gehrmann et al. (2019) and ranks tokens based on their model-assigned probabilities. All detectors are tested under the same black-box constraints using 60 paraphrased variants per input to ensure fairness.

## 5.2 EMPIRICAL STUDY ON HYPERPARAMETERS

The implementation of the proposed **GramGuard** involves the setting of two hyperparameters: (1) the N-Gram ensemble, which controls the diverse combination of the Grams from 2 to 5; (2) the temperature of the source model - $q_\tau$ that directs the process of data perturbation for variants production. In other words, a higher temperature leads to a smoother distribution of tokens' probabilities compared with that of the lower ones. In this group of experiments, we aim to empirically confirm the values of the two hyperparameters that can yield the best performance. Fig. 3 illustrates ROC-AUC performance of the top-5 N-Gram combinations (i.e., 2+3+4+5 Grams, 2+3+4 Grams, 2+3+5 Grams, 3+4+5 Grams, and 2+4 Grams) across the nine datasets by sweeping decoding temperature $\tau \in \{0.1, 0.3, 0.5, 0.7, 0.9, 1.1\}$. Other possible combinations of N-Grams are not shown in the Fig. 3 because they yield worse performance than the top-5 ones.

Table 1: The ROC-AUC comparison among SOTA baselines and GramGuard across three datasets (PubMed, Writing, XSum) and three paraphrasing sources (Gemini, GPT-3.5-Turbo, GPT-4) under the settings of 2+3+4+5 Grams and 0.1 sampling temperature. Bolded values indicate the best performance per source-specified dataset.

| Models | Gemini | | | | GPT-3.5-Turbo | | | | GPT-4 | | | |
|---|---|---|---|---|---|---|---|---|---|---|---|---|
| | PubMed | Writing | XSum | AVG | PubMed | Writing | XSum | AVG | PubMed | Writing | XSum | AVG |
| RoBERTa-Base | 0.446 | 0.8002 | 0.8708 | 0.7656 | 0.6188 | 0.7084 | 0.9150 | 0.7474 | 0.5309 | 0.5068 | 0.6778 | 0.5718 |
| RoBERTa-Large | 0.4508 | 0.6296 | 0.8101 | 0.6301 | 0.5480 | 0.8507 | 0.8507 | 0.7915 | 0.6067 | 0.3821 | 0.6879 | 0.5589 |
| GPTZero | 0.884 | **0.9837** | 0.9987 | 0.9554 | 0.8799 | 0.9292 | 0.9952 | 0.9347 | 0.8482 | 0.8262 | 0.9815 | 0.8853 |
| Likelihood | 0.7616 | 0.9114 | 0.8519 | 0.8416 | 0.8775 | 0.9740 | 0.9578 | 0.9364 | 0.7980 | 0.8553 | 0.8104 | 0.8212 |
| Entropy | 0.4335 | 0.4395 | 0.5399 | 0.4709 | 0.2767 | 0.1902 | 0.3305 | 0.2658 | 0.3295 | 0.3702 | 0.4360 | 0.3786 |
| LogRank | 0.7689 | 0.9076 | 0.8628 | 0.8464 | 0.8687 | 0.9656 | 0.9582 | 0.9308 | 0.8003 | 0.8286 | 0.7975 | 0.8088 |
| LRR | 0.7234 | 0.9179 | 0.7274 | 0.7562 | 0.7433 | 0.8958 | 0.9162 | 0.8517 | 0.6814 | 0.7028 | 0.7447 | 0.7093 |
| NPR | 0.6384 | 0.9487 | 0.8172 | 0.8014 | 0.6784 | 0.8924 | 0.7899 | 0.7869 | 0.6328 | 0.6122 | 0.5280 | 0.591 |
| DNAGPT | 0.5199 | 0.9257 | 0.8675 | 0.7710 | 0.7959 | 0.9425 | 0.9124 | 0.8836 | 0.7565 | 0.8032 | 0.7347 | 0.7648 |
| DetectGPT | 0.6854 | 0.9151 | 0.7549 | 0.7851 | 0.7444 | 0.8811 | 0.8416 | 0.8223 | 0.6805 | 0.6217 | 0.5660 | 0.6227 |
| Fast-DetectGPT | 0.8769 | 0.9465 | 0.8518 | 0.8917 | 0.9021 | **0.9916** | 0.9907 | 0.9614 | 0.8503 | 0.9612 | 0.9067 | 0.9248 |
| **GramGuard** | **0.8879** | 0.9803 | **0.9992** | **0.9558** | **0.9616** | 0.9860 | **0.9962** | **0.9812** | **0.9508** | **0.9731** | **0.9990** | **0.9743** |
| *(Absolute ↑)* | *0.39%* | *-0.34%* | *0.05%* | *0.04%* | *5.95%* | *-0.56%* | *0.1%* | *1.98%* | *10.05%* | *1.19%* | *1.75%* | *4.95%* |

We observed a relatively downward trend as the decoding temperature $\tau$ increases. This trend reflects how temperature governs the paraphrastic entropy of generated variants. At lower temperatures, perturbations remain closer to the original phrasing, allowing our detector to reliably capture subtle statistical shifts introduced by machine text. In contrast, higher temperatures yield more randomized and human-like outputs, which can obscure underlying machine patterns, thereby degrading detection performance. Nevertheless, GramGuard consistently achieves AUCs above 0.93 even under these high-entropy settings, which demonstrates strong resilience. It can also be observed from Fig. 3 that the 2+3+4+5 gram ensemble at $\tau = 0.1$ yields the relatively highest and most stable performance across all the datasets. This setup captures both short-range and long-range n-gram fluency deviations while leveraging the lowest-entropy paraphrases, maximizing sensitivity to stylistic distortions that differentiate human and AI text. We therefore adopt this combination as the default configuration for all subsequent evaluations.

## 5.3 Performance Comparison with SOTA Detectors

To emphasize the superiority of GramGuard over other detection methods in identifying machine-generated text, we present the following two key observations from the results demonstrated in the Table 1.

First, GramGuard consistently outperforms all baselines across diverse datasets and generative sources, achieving the highest average ROC-AUC scores. Crucially, GramGuard exhibits exceptional robustness against paraphrasing attacks, which is a key weakness of prior methods. While zero-shot curvature detectors like DetectGPT and Fast-DetectGPT degrade significantly under adversarial rewriting, GramGuard maintains near-perfect separability (e.g., 0.9508 on PubMed with GPT-4). Similarly, probability-based scorers (LogRank, LRR) and N-Gram-based DNAGPT show instability, especially on complex datasets like XSum, whereas GramGuard sustains AUCs > 0.99. This advantage stems from its core design that leverages pre-LLM n-gram models as a statistically pure baseline and measures delta features (log-likelihood, entropy, variance shifts) across paraphrased variants to expose the rigidity of AI text under perturbation.

Second, GramGuard's dominance is most pronounced against other baselines under high-fluency source models like *GPT-4*, where it surpasses all competitors by significant margins. For example, GramGuard outperforms Fast-DetectGPT by 10.05% AUC on PubMed. This highlights its superior generalization in black-box settings, which is a challenge for supervised classifiers and entropy-based methods. Critically, unlike other approaches, GramGuard achieves SOTA results using three interpretable delta features coupled with multi-granular n-gram signals verified on a lightweight XGBoost classifier. It is noteworthy that the fusion of multi-granular n-grams can capture nuanced statistical deviations impervious to paraphrasing, as evidenced by its sustained high performance even under high-temperature perturbations (See Fig. 3). Furthermore, the impact of message length in system performance has been evaluated and demonstrated in Appendix D.

| | Only $\mathcal{L}$ | Only $\mathcal{H}$ | Only $\mathcal{V}$ | $\mathcal{L} + \mathcal{H}$ | $\mathcal{L} + \mathcal{V}$ | $\mathcal{H} + \mathcal{V}$ | All Three |
|---|---|---|---|---|---|---|---|
| PubMed \| GPT-4 | 0.628 | 0.905 | 0.962 | 0.922 | 0.972 | 0.975 | 0.977 |
| PubMed \| GPT-3.5-Turbo | 0.690 | 0.899 | 0.968 | 0.926 | 0.975 | 0.977 | 0.978 |
| PubMed \| Gemini | 0.673 | 0.861 | 0.884 | 0.875 | 0.910 | 0.934 | 0.943 |
| Writing \| GPT-4 | 0.747 | 0.845 | 0.931 | 0.878 | 0.954 | 0.955 | 0.959 |
| Writing \| GPT-3.5-Turbo | 0.891 | 0.914 | 0.947 | 0.934 | 0.980 | 0.980 | 0.980 |
| Writing \| Gemini | 0.729 | 0.784 | 0.884 | 0.956 | 0.940 | 0.936 | 0.982 |
| XSum \| GPT-4 | 0.915 | 0.945 | 0.973 | 0.962 | 0.993 | 0.993 | 0.994 |
| XSum \| GPT-3.5-Turbo | 0.945 | 0.970 | 0.986 | 0.981 | 0.993 | 0.994 | 0.993 |
| XSum \| Gemini | 0.819 | 0.907 | 0.966 | 0.995 | 0.982 | 0.987 | 0.999 |

Figure 4: Ablation heatmap showing ROC–AUC for seven feature sets (Only $\mathcal{L}$, Only $\mathcal{H}$, Only $\mathcal{V}$, $\mathcal{L} + \mathcal{H}$, $\mathcal{L} + \mathcal{V}$, $\mathcal{H} + \mathcal{V}$, All Three) across the nine dataset–LLM variants. Bolder cells indicate higher AUC scores.

## 5.4 FEATURE ABLATION: ENTROPY, LIKELIHOOD, FREQUENCY

To further investigate the role of the three N-Gram features (i.e., $\mathcal{L}$: log-likelihood, $\mathcal{H}$: Entropy, and $\mathcal{V}$: statistical variance) in impacting the performance of AI text detection, we conduct an ablation study by disabling each of the features as shown in Fig 4. It can be observed that $\Delta_{\mathcal{L}}$-only exhibits the weakest performance across all dataset-LLM pairs. This occurs because $\Delta_{\mathcal{L}}$ primarily measures *fluency deviations* after paraphrasing, which means human text shows significant probability drops due to lexical creativity, but machine-generated text maintains rigid phrasing with minimal likelihood shifts. However, this metric proves sensitive to sparse paraphrases, limiting its standalone reliability. Conversely, $\Delta_{\mathcal{H}}$ and $\Delta_{\mathcal{V}}$ demonstrate superior robustness. $\Delta_{\mathcal{H}}$ quantifies *lexical unpredictability* because of human rewrites increasing entropy through diverse word choices, whereas AI text displays distributional brittleness with negligible entropy changes. $\Delta_{\mathcal{V}}$ captures *repetition rigidity* via n-gram frequency dispersion that machine text resists variance shifts under perturbation, while human writing exhibits flexible redistribution. On the other hand, the $\Delta_{\mathcal{H}} + \Delta_{\mathcal{V}}$ combination nearly matches full-triad performance, indicating these metrics are primary discriminators. They reveal the *stylistic rigidity* of LLM content that synthetic text fails to mimic human lexical diversity and dynamic phrasing, even when paraphrased. $\Delta_{\mathcal{L}}$ provides an auxiliary signal by contextualizing entropy/variance shifts within probabilistic coherence. The synergy of the three features enables robustness against adversarial perturbations by collectively unmasking the statistical homogeneity of machine-generated text. More details about the impact of the three features on system performance across various N-Gram combinations can be found in Appendix B and temperature-wise results in Appendix C (See the Supplementary Materials).

## 6 CONCLUSION

This study initially proposes to leverage pre-AI N-Gram models exclusively trained on human corpora as the "gold standard" for AI text detection. Building on this foundation, we introduce GramGuard, a lightweight, interpretable framework that identifies machine-generated text through delta-based statistical analysis of paraphrastic variants. Our core innovation lies in measuring systematic shifts in three key metrics: log-likelihood, entropy, and token frequency variance across perturbations generated by LLMs under controlled temperatures. We found that AI-generated texts exhibit smaller and inconsistent deviations compared to human-authored content. Extensive experiments across PubMed, WritingPrompts, and XSum datasets demonstrate that GramGuard achieves significant ROC-AUC and exceptional robustness against paraphrasing attacks compared with various SOTA baselines. Ablation studies confirm that entropy and frequency variance deltas are primary discriminators, revealing AI text's inherent lexical inflexibility. Future work will explore hybrid approaches combining token-level robustness with corpus-level interpretability.

## ETHICS STATEMENT

This research adheres to the ethical standards of the ICLR community. All datasets used are publicly available. No personally identifiable or sensitive data was included, and the experiments do not encourage bias, discrimination, or unsafe applications.

## REPRODUCIBILITY STATEMENT

We provide comprehensive implementation details to ensure reproducibility. All datasets are publicly accessible, and the $n$-gram models are released in Appendix E along with download instructions. The whole together, these resources enable independent replication of all results.

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

## A  APPENDIX

### A.1  APPENDIX A: PROMPT TEMPLATES FOR PARAPHRASING

To generate paraphrased variants for robustness evaluation, we employed ChatGPT APIs with a consistent function-calling interface. Each input sentence was rewritten into 10 distinct variants per temperature, across six decoding temperatures ($T \in \{0.1, 0.3, 0.5, 0.7, 0.9, 1.1\}$). This setup yielded 60 paraphrases per model per sentence, supporting extensive evaluation of detection performance under stylistic variation.

#### FUNCTION DEFINITION: REWRITE_SENTENCE()

The same structured function definition was used across all APIs:

```
{
  "name": "rewrite_sentence",
  "description": "Rewrites a given sentence while preserving the original
                  meaning. The output should be fluent and natural.",
  "parameters": {
    "type": "object",
    "properties": {
      "sentence": {
        "type": "string",
        "description": "The input sentence to paraphrase"
      }
    },
    "required": ["sentence"]
  }
}
```

#### CHATGPT (GPT-4.1-MINI) INVOCATION

Each paraphrase was generated using OpenAI's function-calling API as follows:

```
response = client.chat.completions.create(
    model="gpt-4.1-mini",
    temperature=T,
    messages=[
        {"role": "system",
         "content": "You rewrite text fluently and clearly."},
        {"role": "user",
         "content": "Rewrite the following sentence while preserving its "
                    "meaning:\n\n\"{sentence}\""}
    ],
    functions=[rewrite_sentence],
    function_call={"name": "rewrite_sentence"}
)
```

#### VARIANT GENERATION NOTES

- A total of 60 variants per sentence were created (6 temperatures × 10 samples).
- Each variant was stored in structured CSVs under columns `variant_1` to `variant_10`.

## Appendix B: Feature Ablation and Overfit Gap Analysis

To detail the impact of each delta feature, we performed a feature-ablation study over all combinations of 2-, 3-, 4-, and 5-gram statistics (log-score, entropy, variance) across datasets, each under varied decoding temperatures.

For every feature set, we recorded:

- Test accuracy and ROC-AUC

- 5-fold CV AUC (mean ± std)

- Overfit gap (Test AUC − CV AUC mean)

Observed trends:

- The full 2+3+4+5-gram combination attained the best balance of AUC and stability.

- Employing only 4-gram and 5-gram features markedly reduced performance and widened the overfit gap.

- Adding higher-order n-grams to a 2+3-gram backbone yields modest AUC gains at the cost of slight instability.

- The complete 12-dimensional delta vector (log-score, entropy, variance × each of the four n-gram orders) outperformed any single-metric subset.

Table 2: Feature ablation results on PubMed | GPT-4. Top five model combinations.

| Feature Set | Test AUC | CV AUC Mean | Gap |
|---|---|---|---|
| 2+3+4+5-gram | **0.9734** | **0.9739** | **0.0004** |
| 3+4-gram | 0.9610 | 0.9587 | 0.0023 |
| 4+5-gram | 0.9448 | 0.9433 | 0.0015 |
| 2-gram only | 0.9444 | 0.9345 | 0.0099 |
| 5-gram only | 0.9175 | 0.9170 | 0.0005 |

**(PubMed | GPT-4).** The 2+3+4+5-gram feature set achieves the highest Test AUC (0.9734) with a negligible overfit gap (0.0004), demonstrating both accuracy and stability. Medium-order combinations (3+4-gram) and single-order subsets (e.g., 2-gram only) show lower AUC and/or larger gaps.

Table 3: Feature ablation results on PubMed | Gemini. Top five model combinations.

| Feature Set | Test AUC | CV AUC Mean | Gap |
|---|---|---|---|
| 2+3+4+5-gram | **0.9496** | **0.9422** | **0.0074** |
| 2-gram only | 0.9247 | 0.9080 | 0.0168 |
| 3+4-gram | 0.9082 | 0.9026 | 0.0056 |
| 4+5-gram | 0.8906 | 0.8826 | 0.0079 |
| 5-gram only | 0.8618 | 0.8503 | 0.0115 |

**(PubMed | Gemini).** The full vector again tops performance (Test AUC 0.9496, gap 0.0074). Mid/single-order subsets fall in both AUC and stability, underscoring the importance of multi-scale n-gram deltas.

Table 4: Feature ablation results on PubMed | GPT-3.5-Turbo. Top five model combinations.

| Feature Set | Test AUC | CV AUC Mean | Gap |
|---|---|---|---|
| 2+3+4+5-gram | **0.9779** | **0.9772** | **0.0007** |
| 3+4-gram | 0.9633 | 0.9626 | 0.0007 |
| 2-gram only | 0.9512 | 0.9449 | 0.0063 |
| 4+5-gram | 0.9441 | 0.9438 | 0.0003 |
| 5-gram only | 0.9156 | 0.9102 | 0.0054 |

**(PubMed | Turbo 3.5).** PubMed | Turbo 3.5 variant's yield the highest overall AUC (0.9779) and the smallest gap (0.0007) for the full feature set, indicating exceptional robustness to paraphrasing. Even here, reduced feature combinations lead to noticeable drops in performance or increased overfitting, reaffirming that the 12-dimensional delta representation is essential for reliable detection on PubMed.

Table 5: Feature ablation results on Writing | GPT-4. Top five model combinations.

| Feature Set | Test AUC | CV AUC Mean | Gap |
|---|---|---|---|
| 2+3+4+5-Gram | **0.9549** | **0.9491** | **0.0058** |
| 3+4-Gram | 0.9292 | 0.9228 | 0.0064 |
| 2-Gram only | 0.9122 | 0.9019 | 0.0103 |
| 4+5-Gram | 0.9109 | 0.9048 | 0.0061 |
| 5-Gram only | 0.8996 | 0.8890 | 0.0106 |

**(Writing | GPT-4).** Full 2+3+4+5-gram feature set achieves the highest Test AUC (0.9549) with a modest overfit gap (0.0058). Mid-order combinations (3+4-gram) and single-order subsets (2-gram only, 5-gram only) show noticeably lower AUCs and larger gaps, indicating reduced stability and generalization.

Table 6: Feature ablation results on Writing | Gemini. Top five model combinations.

| Feature Set | Test AUC | CV AUC Mean | Gap |
|---|---|---|---|
| 2+3+4+5-Gram | **0.9784** | **0.9762** | **0.0022** |
| 3+4-Gram | 0.9752 | 0.9728 | 0.0024 |
| 4+5-Gram | 0.9663 | 0.9652 | 0.0011 |
| 5-Gram only | 0.9643 | 0.9637 | 0.0006 |
| 2-Gram only | 0.9564 | 0.9493 | 0.0071 |

**(Writing | Gemini).** The combined 2+3+4+5-gram delta vector yields a Test AUC of 0.9784 and a minimal gap of 0.0022, outperforming all reduced subsets. The substantially larger gaps for 2-gram only (0.0071) and 4+5-gram (0.0011) confirm that multi-order integration is critical to maintain both high accuracy and low generalization error on creative text.

Table 7: Feature ablation results on Writing | GPT-3.5-Turbo. Top five model combinations.

| Feature Set | Test AUC | CV AUC Mean | Gap |
|---|---|---|---|
| 2+3+4+5-Gram | **0.9798** | **0.9749** | **0.0049** |
| 3+4-Gram | 0.9689 | 0.9638 | 0.0051 |
| 4+5-Gram | 0.9616 | 0.9519 | 0.0097 |
| 2-Gram only | 0.9578 | 0.9491 | 0.0087 |
| 5-Gram only | 0.9553 | 0.9457 | 0.0097 |

**(Writing | Turbo 3.5).** The full 2+3+4+5-gram set again tops performance with 0.9798 AUC and a gap of 0.0049. Reduced feature sets exhibit lower AUCs and increased overfitting (e.g., 4+5-gram

gap of 0.0097), underscoring that the complete delta representation is essential for robust detection on writing-style data.

Table 8: Feature ablation results on XSum | GPT-4. Top five model combinations.

| Feature Set | Test AUC | CV AUC Mean | Gap |
|---|---|---|---|
| 2+3+4+5-Gram | **0.9900** | **0.9877** | **0.0023** |
| 3+4-Gram | 0.9802 | 0.9772 | 0.0030 |
| 2-Gram only | 0.9770 | 0.9738 | 0.0032 |
| 4+5-Gram | 0.9734 | 0.9689 | 0.0045 |
| 5-Gram only | 0.9707 | 0.9666 | 0.0041 |

**(XSum | GPT-4).** Full 2+3+4+5-gram set achieves an outstanding Test AUC of 0.9900 with a minimal overfit gap of 0.0023. Reduced combinations—such as 3+4-gram (AUC 0.9802, gap 0.0030) or 2-gram only (AUC 0.9770, gap 0.0032)—display slightly lower accuracy and marginally larger gaps, demonstrating that integrating all four n-gram orders is key for both performance and stability.

Table 9: Feature ablation results on XSum | Gemini. Top five model combinations.

| Feature Set | Test AUC | CV AUC Mean | Gap |
|---|---|---|---|
| 2+3+4+5-Gram | **0.9989** | **0.9990** | **0.0001** |
| 3+4-Gram | 0.9986 | 0.9986 | 0.0000 |
| 4+5-Gram | 0.9970 | 0.9968 | 0.0002 |
| 5-Gram only | 0.9965 | 0.9960 | 0.0005 |
| 2-Gram only | 0.9919 | 0.9908 | 0.0011 |

**(XSum | Gemini).** Combined 2+3+4+5-gram vector reaches near-perfect cross-validation generalization (CV AUC 0.9990) and a negligible gap (0.0001), yielding a Test AUC of 0.9989. Even mid-order subsets (e.g., 3+4-gram with gap 0.0000) remain strong, but none match the consistency and peak accuracy of the full delta representation.

Table 10: Feature ablation results on XSum | GPT-3.5-Turbo. Top five model combinations.

| Feature Set | Test AUC | CV AUC Mean | Gap |
|---|---|---|---|
| 2+3+4+5-Gram | **0.9954** | **0.9955** | **0.0001** |
| 3+4-Gram | 0.9923 | 0.9927 | 0.0004 |
| 2-Gram only | 0.9911 | 0.9912 | 0.0001 |
| 4+5-Gram | 0.9880 | 0.9878 | 0.0002 |
| 5-Gram only | 0.9848 | 0.9848 | 0.0001 |

**(XSum | Turbo 3.5)** Full feature set again dominates, posting a Test AUC of 0.9954 and an almost zero gap (0.0001). The tiny performance drop in reduced sets (e.g., 4+5-gram gap 0.0002) highlights how the 12-dimensional delta features reliably generalize even on highly abstractive summarization data.

APPENDIX C: FULL TEMPERATURE-WISE DETECTION RESULTS

This appendix details the ROC-AUC performance of our delta-feature detector (using an XGBoost classifier) across decoding temperatures $\tau = [0.1, 0.3, 0.5, 0.7, 0.9, 1.1]$ for three ChatGPT variants (GPT-4, GPT-3.5-Turbo, Gemini) on PubMed abstracts, WritingPrompts passages, and XSum summaries.

Overall, all variants maintain high discriminative power (AUC > 0.94) at every $\tau$, yet each shows characteristic strengths and sensitivities:

- **GPT-4:** Reaches its highest PubMed AUC (0.9577) at $\tau = 0.7$ and peaks on XSum (0.9992) already at $\tau = 0.1$. WritingPrompts performance is strongest at $\tau = 0.1$ (0.9712), dipping slightly between $\tau = 0.5$ and $\tau = 0.9$, indicating that mid-range sampling smoothness introduces modest variability in both formal and creative prose.

- **GPT-3.5-Turbo:** Exhibits robust stability across all $\tau$, with PubMed and XSum AUCs peaking at higher settings (PubMed: 0.9674 at $\tau = 1.1$; XSum: 0.9981 at $\tau = 0.5$). Writing-Prompts detection is best at $\tau = 0.1$ (0.9864). Fluctuations remain within 0.03 AUC, showing low sensitivity to decoding randomness.

- **Gemini:** Delivers near-ceiling XSum AUCs ($\geq 0.9951$), but underperforms on PubMed ($\approx 0.87$–$0.90$), with a trough at $\tau = 0.7$ (0.8799). WritingPrompts stays strong (0.95–0.98), peaking at $\tau = 0.5$ (0.9757). This pattern highlights Gemini's relative difficulty detecting biomedical paraphrases under moderate randomness.

GPT-4 and GPT-3.5-Turbo achieve marginally higher and more consistent AUCs on formal texts (PubMed, XSum), whereas Gemini's only notable weakness is on PubMed at mid-range $\tau$.

Table 11: ROC-AUC at decoding temperatures $\tau$ for paraphrasers (XGB classifier) on PubMed, WritingPrompts, and XSum.

| $\tau$ | GPT-4.1-mini | | | GPT-3.5-Turbo | | | Gemini | | |
|---|---|---|---|---|---|---|---|---|---|
| | PubMed | Writing | XSum | PubMed | Writing | XSum | PubMed | Writing | XSum |
| 0.1 | 0.9460 | 0.9712 | 0.9992 | 0.9627 | 0.9864 | 0.9952 | 0.8727 | 0.9725 | 0.9981 |
| 0.3 | 0.9494 | 0.9405 | 0.9915 | 0.9567 | 0.9859 | 0.9950 | 0.8909 | 0.9628 | 0.9984 |
| 0.5 | 0.9409 | 0.9471 | 0.9924 | 0.9508 | 0.9824 | 0.9981 | 0.9033 | 0.9757 | 0.9994 |
| 0.7 | 0.9577 | 0.9308 | 0.9878 | 0.9476 | 0.9731 | 0.9971 | 0.8799 | 0.9767 | 0.9951 |
| 0.9 | 0.9466 | 0.9348 | 0.9844 | 0.9526 | 0.9734 | 0.9949 | 0.8900 | 0.9733 | 0.9984 |
| 1.1 | 0.9458 | 0.9532 | 0.9843 | 0.9674 | 0.9717 | 0.9931 | 0.8905 | 0.9508 | 0.9977 |

## APPENDIX D: LENGTH-ROBUSTNESS ANALYSIS

To understand how detection scales with input length, we truncated each dataset to 45, 90, 135, and 180 words and re-evaluated our XGBoost models (using the full 12-dimensional delta vector) for GPT-4.1-mini, Gemini, and GPT-3.5-Turbo. For each truncated set, we binned examples by word count, computed mean length and ROC-AUC per bin, and plotted the results in Figure 5.

- **XSum (Fig. 5a):** Even at 45 words, AUC ≈ 0.97 across all variants, rising above 0.99 by 90 words and then leveling off.
- **WritingPrompts (Fig. 5b):** AUC climbs from ~0.80 at 45 words to ~0.96 at 120 words for GPT-4.1-mini and GPT-3.5-Turbo; Gemini dips by ~0.02 at 180 words.
- **PubMed (Fig. 5c):** AUC exceeds 0.85 at 45 words and reaches > 0.95 by 60 words, then flattens.

These findings confirm that longer passages generally improve discriminability, yet our delta-feature detector remains robust even on very short inputs, with each dataset exhibiting its own length-sensitivity profile.

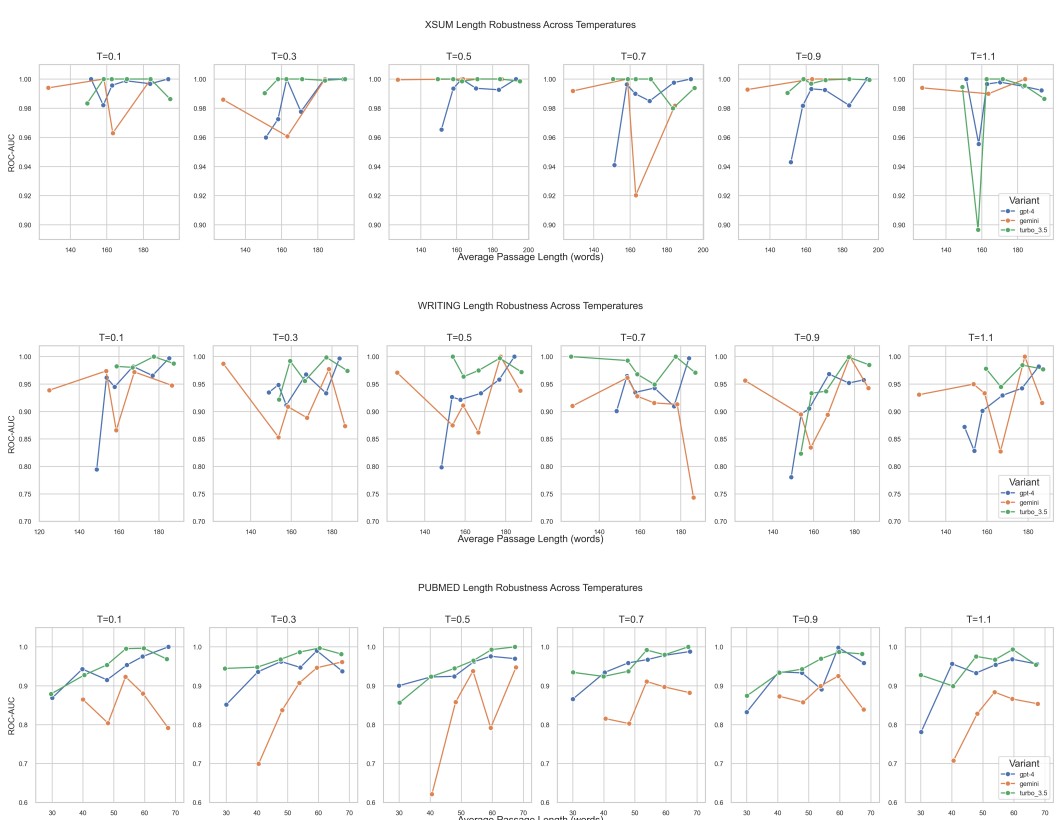

Figure 5: Length-robustness of delta-feature detector across temperatures and datasets.

## APPENDIX E: PRE-TRAINED MODEL DOWNLOAD AND SETUP

To replicate our detection pipeline, we provide four pre-trained $n$-gram language models (2-gram to 5-gram) on Hugging Face[2]. These KenLM binaries were trained on a large, clean corpus of human-written English text. They serve as stable, interpretable statistical baselines for detecting linguistic perturbations introduced by large language models. By comparing n-gram statistics before and after paraphrasing, our approach quantifies how AI-generated text diverges from human norms. These models are integral to computing delta log-likelihoods, entropy, and frequency variance features used throughout our detection pipeline.

Please download and place the following files into a directory named `models/` at the root of the project:

- `2-gram.arpa.bin`
- `3-gram.arpa.bin`
- `4-gram.arpa.bin`
- `5-gram.arpa.bin`

After downloading, the directory structure should look like this:

```
Ngram_DetectGPT/
|- models/
|- 2-gram.arpa.bin
|- 3-gram.arpa.bin
|- 4-gram.arpa.bin
|- 5-gram.arpa.bin
```

Our detection scripts will automatically load these models to compute n-gram log-likelihood, entropy, and frequency-based delta features.The full implementation and instructions are available on GitHub[3].

---

[2]`https://huggingface.co/NGramDev/ngram-detect-models`
[3]`https://github.com/N-Gram-dev/GramGuard`

## APPENDIX F: USE OF LLMS

In accordance with the ICLR 2026 submission policy, we disclose the use of large language models (LLMs). LLMs were used for:

- Generating paraphrased text variants for robustness experiments;
- Drafting and refining some sentences for the introduction and related work, which were subsequently revised by the authors;
- Proofreading and minor formatting adjustments.

All methodological designs, theoretical derivations, experiments, and analyses are the original work of the authors.

