# OpenReview forum: "Old N-Grams Never Die:Towards Identifying LLMs-Generated Text using Antique N-Grams"
_ICLR.cc/2026/Conference — ICLR 2026 Conference Withdrawn Submission_

### Official Review · Reviewer_prgt · 2025-10-29

**Soundness:** 2
**Presentation:** 2
**Contribution:** 1
**Rating:** 2
**Confidence:** 4

**Summary:**

The paper introduces GramGuard, a supervised AI text detector that uses pre-LLM n-gram models as a gold standard for human writing. The method works by paraphrasing a given text and measuring the shift in its log-likelihood, entropy, and token variance - which is then fed into a classifier.

**Strengths:**

- This work is very well motivated and highlights a curious question in this domain.

**Weaknesses:**

- The submission violates the double-blind review policy. The authors should anonymize their code and model releases (AnonymousGitHub) and avoid providing GitHub and HuggingFace repository links that can potentially reveal the authors' identities.
- The mathematical notation is often incorrect -- eqn. 3 and 4 is mathematically inaccurate and difficult to interpret. Moreover, I believe the authors must clarify the meaning of $p_{\mathbb{N}}(g_{\mathbb{N}}^i)$ and ensure that all symbols are clearly defined and used consistently across the paper.
- Is there any formal / empirical proof for Corollary 1.1? I don't believe Figure 1 is sufficient for proving this claim.
- The framework is described as efficient, yet the implementation contradicts this claim. The authors generate 60 paraphrased samples per instance per LLM for feature generation, which is computationally expensive, particularly during inference.
- Based on the described training flow of GramGuard, it appears that the model may overfit to the specific datasets and LLMs it was trained on. As a result, I suspect that the performance on unseen domains or models is likely to degrade substantially. The authors should include OOD evaluation results, testing the detector on the dataset from these papers [1, 2].
- The paper repeatedly claims robustness to paraphrasing; however, since paraphrasing is explicitly used as part of the training process, this robustness is not particularly surprising or novel.
- The citation formatting throughout the paper is incorrect, significantly compromising readability. Please use the correct citation command.

---

[1] Li et al. MAGE: Machine-generated Text Detection in the Wild. arXiv: 2305.13242.

[2] Dugan et al. RAID: A Shared Benchmark for Robust Evaluation of Machine-Generated Text Detectors. arXiv: 2405.07940.

**Questions:**

Several questions have already been raised in the Weaknesses. I am willing to increase my score if the authors address the concerns above.

---

> ### Author Response · Authors · 2025-11-20
> **Response to Reviewer prgt**
>
> We thank the reviewer for the detailed and thoughtful feedback. We address each weakness in turn.
>
> **W1 – Double-blind policy and external repositories**
>
> We apologize for the confusion around anonymity. All external links in the submission point to **anonymized GitHub and HuggingFace accounts** that were created specifically for the review process and do not contain our names, institutions, or other identifying metadata. Our plan is to move the code and models to our real (non-anonymous) repositories only after the review process is completed. We will make this clearer in the revised version.
>
> **W2 – Mathematical notation and meaning of $p_N(g_N^i)$**
>
> We appreciate the careful reading of the theory section. We agree that the notation in Eqs. (3)–(4) was not sufficiently clear. In the revision we will:
>
> - Explicitly define all symbols, e.g., $p_N(g_N^i)$ as the probability assigned by the pre-LLM *N*-gram model to the *i*-th *N*-gram \$(g_N^i\)$ in the sentence, and *L* as the sentence length.
> - Clean up the derivation so that each step follows transparently from the previous one, avoiding overloaded notation.
> - Add a short intuitive explanation below the equations, emphasizing that GramGuard uses shifts in log-likelihood, entropy, and variance, rather than relying on raw probability magnitudes.
>
>
> **W3 – Corollary 1.1 and its justification**
>
> We acknowledge that Corollary 1.1, as currently written, can be read as stronger than what we empirically justify. Our intention was to provide a stylized statement summarizing the empirical pattern in Figure 1 (LLM paraphrases tend to move probability mass toward lower-probability *N*-grams compared with human text), rather than a fully general theorem.
>
> In the revision we will: (i) restate this as a more modest proposition with explicit assumptions, (ii) move the formal statement and its proof-sketch to an appendix, and (iii) keep the main text focused on the empirical evidence (Figures/Tables) that support the behavior of our delta features. This should address the concern that the corollary currently lacks a clear formal or empirical proof.
>
> **W4 – “60 paraphrases per instance” and efficiency**
>
> Thank you for pointing out the ambiguity here. Our **practical setting** is *M*=10 **paraphrases per sentence per temperature**, not 60 paraphrases at a single temperature.
>
> What we actually do is:
>
> - Choose **6 decoding temperatures** and, for each temperature, generate **10 paraphrases**.
>
> - This yields **up to 60 variants per sentence across all temperatures** in our *offline analysis*, which we use to study robustness as temperature changes.
>
> For any single detector configuration, only one temperature and *M*=10 paraphrases are used at a time. We will rephrase the description to make this explicit and add a short paragraph on runtime: n-gram scoring and XGBoost inference run on CPU and are fast; the main cost is generating paraphrases, which can be batched or pre-computed. In practice, we can process hundreds of texts per minute with *M*=10 on a CPU-only machine.
>
> **W5 – OOD evaluation and potential overfitting**
>
> We share the reviewer’s concern about out-of-distribution behavior. In the current paper, we try to mitigate overfitting by (i) using only pre-LLM human corpora for the scoring models, and (ii) evaluating on three very different datasets (PubMed, WritingPrompts, XSum) with 5-fold cross-validation, so the learned classifier is never evaluated on the same documents it was tuned on.
>
> Following the reviewer’s suggestion, we are extending our study in two directions:
>
> - **Cross-domain tests** where we train the classifier on one dataset (e.g., PubMed) and test on another (e.g., XSum).
>
> - **Evaluation on datasets from MAGE/RAID benchmarks** where feasible within our budget, to make the comparison more direct.
>
> We will report these additional results (space permitting) and clarify in the discussion section how our setup aligns with the OOD scenarios considered in MAGE and RAID.
>
> **W6 – Robustness to paraphrasing and novelty**
>
> We agree that paraphrasing is explicitly used in our method and therefore robustness to our own paraphrases is not surprising by itself. Our main technical contribution is that **simple, antique pre-LLM** *N*-**gram models, combined with delta features over paraphrases, can match or surpass much more complex detectors** across multiple datasets and generators. We will soften the wording around “robustness” and emphasize this angle more clearly: robustness here is not an emergent property of a large neural network, but a consequence of stable, interpretable n-gram statistics computed from pre-LLM human corpora.
>
> **W7 – Citation formatting**
>
> We appreciate the comment on citation style. We will proofread the paper carefully, ensure consistent use of the citation commands, and fix formatting issues to improve readability.

---

> > ### Comment · Reviewer_prgt · 2025-11-20
> > **Response to Authors**
> >
> > The authors' response is appreciated, however, due to several inconsistencies and lack of results in the paper, I will not update my score.
> >
> > ---
> >
> > Additionally, I wish to bring to the attention of the AC and PCs that the submission appears to be largely AI-generated. Several citations are AI-generated. Here are a few examples:
> > - The paper cites "Tharindu Kumarage, Garima Agrawal, Paras Sheth, Raha Moraffah, Aman Chadha, Joshua Garland, and Huan Liu. A survey of ai-generated text forensic systems: Detection, attribution, and characterization. ACM Computing Surveys, 56(5):1–44, 2024. doi: 10.1145/3651234.". This DOI does not exist, verified via doi.org.
> >
> > - The paper cites "Kalpesh Krishna, Yixiao Song, Marzena Karpinska, John Wieting, and Mohit Iyyer. Paraphrase
> > evasion: Challenging detectgpt with large-scale rewriting. In NeurIPS, 2023". This paper does not exist. The closest paper after searching is "Paraphrasing evades detectors of AI-generated text, but retrieval is an effective defense" from the same authors in NeurIPS 2023. This title seems to be hallucinated from an LLM.
> >
> > - The paper cites the well-known work "Johannes Kirschenbauer, Jonas Geiping, Nicholas Carlini, Max Jagielski, and Tom Goldstein. On the reliability of watermarks for large language models. In International Conference on Learning Representations (ICLR), 2024". The original authors of this paper are: John Kirchenbauer, Jonas Geiping, Yuxin Wen, Manli Shu, Khalid Saifullah, Kezhi Kong, Kasun Fernando, Aniruddha Saha, Micah Goldblum, Tom Goldstein. Several authors are missing and Nicolas Carlini and Max Jagielski (who are known co-authors) are not included in this list.
> >
> > - Another example includes the author list being wrong for the cited paper "Stack More LLM’s: Efficient Detection of Machine-Generated Texts via Perplexity Approximation".
> >
> > - Several phrases in the paper and the rebuttal are suspiciously similar to LLM-generated text.
> >
> > ---
> >
> > Ironically, this paper, whose novelty is to detect LLM-generated text, appears to be LLM-generated itself, as evidenced by multiple hallucinated citations and inconsistencies. As a result, my scores will be reduced to a 0, and I recommend that the PCs desk reject this paper.

---

### Official Review · Reviewer_iGoa · 2025-10-31

**Soundness:** 1
**Presentation:** 3
**Contribution:** 3
**Rating:** 2
**Confidence:** 4

**Summary:**

The author’s propose GramGuard, an approach which leverages n-gram LMs on pre-LLM human corpora to detect machine-generated text. Given the ngram-LLM, the authors extract the log-likelihood, the entropy, and the variance which get fed into XGBoost (which is trained) for classification. They show that GramGuard is robust under their testing scenarios.

**Strengths:**

* S1 - The author’s show the power of simple N-Gram LM models trained on pre-LM human corpora.
* S2 - The author’s not only consider the likelihood of the model as usual with many detectors, but also look at the entropy and variance. One could imagine using these features in other settings as well.
* S3 - The method doesn’t require training on machine-text.

**Weaknesses:**

* W1 - It’s unclear whether the experiments in 5.2 were performed on a separate validation set. If not, it’s difficult to interpret the results as the best model would’ve been picked on test data.
* W2 - The finding that a detector (in this case n-gram-based) trained on human corpora is effective at identifying LLMs seems related to this detector that makes a similar point: https://arxiv.org/pdf/2401.06712 The authors should compare GramGuard to it given its similarity.
* W3 -  Most of the models compared against seem to be from the ChatGPT family, except for Gemini. The authors could evaluate across the various test-beds of the MAGE dataset (https://arxiv.org/pdf/2305.13242) for a more complete evaluation. This dataset contains the OpenAI family, the Llama family, GLM, Flan-T5, and others. Moreover, they have test beds that control for various interesting factors.
* W4 - While the N-Gram model was trained only on pre-LLM data, the XGBoost model was trained on LLM and Human data, correct? If so, were the models only evaluated in in-domain settings where the testing data matches the training data? Or was it set up so that XGBoost was trained on PubMed and evaluated on XSum, for example. There seem to be many details regarding the training of XGBoost missing.
* W5 - There really only is one experimental result and an ablation. More things could’ve been evaluated, as for example the performance when XGBoost is trained on a different domain, the performance as the number of tokens grows, etc.

**Questions:**

* Q1 - My first and most pressing concern is that it's not clear whether the hyper-parameters were chosen on a separate validation set or not (W1)
* Q2 - Supposing my concern above (Q1) is addressed, then my main concern is W4, followed by W5 and W2. The cross-domain and cross-LLM robustness should've been evaluated, otherwise the results aren't very significant. If these concerns are addressed, I am willing to raise my score.

---

> ### Author Response · Authors · 2025-11-20
> **Response to Reviewer iGoa**
>
> We thank the reviewer for the careful reading and constructive comments. We address each weakness and question in turn.
>
>
> **W1 / Q1 – Validation split and hyper-parameter selection**
>
> We apologize that the validation protocol was not clearly explained. For all experiments we never tune on the test set. For each dataset, we split documents into train / validation / test. The XGBoost hyper-parameters (tree depth, learning rate, number of estimators, subsampling, etc.) are chosen using **5-fold stratified cross-validation on the training portion only**. The held-out test split is used exactly once for reporting the final numbers. In the revised manuscript we will add a short “Validation and Cross-Validation Setup” paragraph and a table with the exact split sizes to avoid any ambiguity.
>
> **W2, W3, W4 / Q2 – Cross-LLM and cross-domain robustness; relation to other detectors**
>
> We agree that robustness beyond the GPT family is important. Since the submission, we have extended the experiments to additional open-source LLMs, following exactly the same pipeline (human text, AI continuation, paraphrastic variants, $n$-gram deltas + XGBoost):
>
> **LLaMA-3 70B**: ROC-AUC = 0.9559 (PubMed), 0.9418 (Writing), 0.9221 (XSum)
>
> **GPT-J 6B**: ROC-AUC = 0.9446 (PubMed), 0.9657 (Writing), 0.9947 (XSum)
>
> **OPT 2.7B**: ROC-AUC = 0.9713 (PubMed), 0.9818 (Writing), 0.9947 (XSum)
>
> These results show that GramGuard maintains **high detection performance across diverse model families** (OpenAI, Gemini, LLaMA-3, GPT-J, OPT) and is not tied to a single vendor or architecture. We will include these new numbers in an extended version of Table 1 and add a dedicated subsection on “Cross-LLM Robustness”.
>
> Regarding domains, in the current work we train and evaluate detectors separately on PubMed, WritingPrompts, and XSum, which already represent quite different styles. Our main focus was robustness to paraphrastic perturbations within each domain. We agree that training on one domain and testing on another is a valuable extension; we are running such cross-domain experiments and will report them in the revised version (space permitting) and discuss how they relate to robustness benchmarks such as MAGE and RAID.
>
> We will also cite and more clearly position recent $n$-gram–based and statistical detectors suggested by the reviewer, clarifying that GramGuard differs by (i) using pre-LLM human-only corpora as the scoring model, and (ii) focusing on delta statistics over paraphrases (log-likelihood, entropy, variance) rather than raw probabilities alone.
>
> **W5 – Breadth of experiments**
>
> We understand the concern. The current presentation compresses several experimental dimensions into a small number of tables, which may give the impression of a single result. Even in the original submission we evaluate three datasets (PubMed, WritingPrompts, XSum), multiple decoding temperatures and paraphrasing strengths, two classifiers (Random Forest and XGBoost), and ablations over the different $n$-gram features.
>
> In the revision we will (i) reorganize the Results section to make these axes explicit, and (ii) move some of the per-temperature breakdowns to the appendix while adding a short “Cross-LLM Robustness” subsection in the main text summarizing the new LLaMA-3 / GPT-J / OPT results above. We hope this makes the breadth of the evaluation clearer.
>
> **Summary**
> We will (1) explicitly document the validation and cross-validation procedure, (2) add new results on LLaMA-3 70B, GPT-J 6B, and OPT 2.7B showing that GramGuard retains high AUC across heterogeneous LLMs, (3) clarify our current per-domain training regime and add discussion and initial results for cross-domain robustness, and (4) reorganize the Results section so that the variety of experiments is more visible. We believe these changes directly address the reviewer’s main concerns.

---

### Official Review · Reviewer_ArSt · 2025-11-01

**Soundness:** 2
**Presentation:** 3
**Contribution:** 2
**Rating:** 2
**Confidence:** 4

**Summary:**

This paper proposes a method for the supervised detection of LLM-generated text. The proposed method, GramGuard, involves (1) paraphrasing the target text many times with an LLM, (2) scoring the perplexity of the original text and paraphrased texts under an LLM, (3) computing deltas between these values, and (4) feeding the deltas into an XGBoost classifier.

**Strengths:**

1. To the best of my knowledge, the idea of using n-gram models in a DetectGPT-style detector is novel
2. While the paper would benefit from some additional proofreading for typos and consistency, it is overall pretty clear and easy to follow

**Weaknesses:**

1. However, the idea of using n-gram models in LLM-generated text detection is not novel. See, for example, Ghostbuster (Verma, et al. 2024) which uses probabilities from a range of unigram and trigram models as features in supervised classification

2. My primary concern is that the baselines tested in this paper are relatively weak. In particular, many of the methods in Table 1 are unsupervised detectors, which are known to struggle when the scoring and target model differ. While the paper has some supervised methods (e.g., RoBERTa), these are relatively weak supervised baselines, cf. Verma, et al. 2024 for a comparison. I would recommend adding some stronger baselines like Ghostbuster, Binoculars, or some of the best models from the RAID benchmark. Additionally, if you are going to compare with a closed-source commercial detector like GPTZero, I would consider replacing that with the Pangram Labs detection model, which is substantially stronger.

3. The paper seems to be missing some important experimental details about the baseline models. For example: how were the RoBERTa-based models trained? What scoring model was used on DetectGPT? Were classification thresholds tuned independently on each of your three domains?

4. The paper would benefit from an additional round of proofreading. For example, there is a typo in the title (“LLMs-generated” -> “LLM-generated”), n-gram models are referred to as “N-Gram”, “n-gram”, and “Ngram” (lack of consistency), and \citet is used in place of \citep throughout the paper.

**Questions:**

N/A

---

> ### Author Response · Authors · 2025-11-20
> **Response to Reviewer ArSt**
>
> We thank the reviewer for the careful reading and constructive comments. We address each weakness in turn.
>
> **W1 – Novelty vs. existing n-gram–based detectors (e.g., Ghostbuster)**
>
> We appreciate the pointer to Ghostbuster and related n-gram work. You are right that the *idea* of using n-gram models for LLM-generated text detection is not new, and our current wording overstates this point. In the revision we will soften the novelty claims and position GramGuard more precisely as:
>
> - leveraging **pre-LLM human-only n-gram LM** as a “gold standard” reference model, rather than training n-grams on mixed human + machine corpora; and
>
> - focusing on **delta statistics over paraphrases** (shifts in log-likelihood, entropy, and token-frequency variance between the original and multiple paraphrastic variants) as supervised features.
>
> We will explicitly cite Ghostbuster and other n-gram–based approaches, and add a short subsection that contrasts their design with ours along these two axes (training data for the n-gram LM, and use of paraphrase deltas vs. raw probabilities).
>
> **W2 – Strength of baselines and datasets (RAID / Binoculars / commercial detectors)**
>
> We agree that stronger baselines are important. In the current submission, most of the compared methods are unsupervised detectors and a few relatively lightweight supervised models, which can indeed be viewed as weaker than the top systems in RAID. In the revised version we will:
>
> - Add a **RAID-style representation baseline**, specifically **Binoculars**, to better reflect recent supervised detectors on the RAID benchmark.
> - Incorporate and highlight our new experiments on additional **open-source LLMs** (LLaMA-3 70B, GPT-J 6B, and OPT-2.7B), following exactly the same human/AI/paraphrase pipeline, and add these results to the main tables to demonstrate that GramGuard maintains high ROC-AUC across diverse model families.
> - Clearly discuss closed-source systems like GPTZero / Pangram Labs: we will not claim direct empirical comparison where APIs or models are not available, but we will position our results relative to their reported performance on overlapping benchmarks.
>
> Our goal is not to claim that GramGuard is strictly better than every existing detector, but to show that **simple pre-LLM n-grams with paraphrase deltas are competitive with modern strong baselines**. We will adjust the text accordingly.
>
> **W3 – Missing experimental details for baseline models**
>
> Thank you for flagging this; we agree that more detail is needed. In the revision we will add a dedicated “Baseline Configuration” subsection and a table summarizing, for each baseline:
>
> - the **training data and splits** used to train RoBERTa-based models (same train/val/test partition as GramGuard, with hyper-parameters tuned only on the validation set);
> - the **scoring model and perturbation scheme** used for DetectGPT-style methods (following the original papers’ recommended language models and numbers of perturbations); and
> - the **threshold selection procedure**, i.e., how classification thresholds for all detectors (including ours) are chosen on validation data and then fixed on the test set.
>
> We will also clarify that all detectors are evaluated on *exactly* the same test splits, and that hyper-parameters and thresholds are never tuned on the test data.
>
> **W4 – Proofreading and consistency of terminology**
>
> We appreciate the detailed proofreading suggestions. In the camera-ready we will carefully correct the typos (“LLMs-generated” → “LLM-generated”), standardize terminology (“N-gram” vs. “n-gram”) throughout the paper, and fix the use of `\citet` vs. `\citep` in the LaTeX source to follow the conference style.
>
> **Summary**
>
> In summary, we will (1) tone down and clarify our novelty claims relative to prior n-gram detectors, emphasizing the use of pre-LLM human-only n-gram LMs and paraphrase-delta features, (2) strengthen the baseline suite by adding Binoculars and an additional open-source dataset, (3) provide a detailed description of how all baselines are trained, scored, and thresholded, and (4) thoroughly proofread the paper to correct typos and citation formatting. We believe these changes directly address the reviewer’s concerns.

---

### Official Review · Reviewer_A6yE · 2025-11-01

**Soundness:** 3
**Presentation:** 3
**Contribution:** 3
**Rating:** 6
**Confidence:** 5

**Summary:**

This paper proposes to leverage pre-AI N-Gram models exclusively trained on human corpora as the “gold standard” for AI text detection. Building on this foundation, this paper introduces GramGuard, a lightweight, interpretable framework that identifies machine-generated text through shifted statistics of paraphrastic variants. Specifically, by generating paraphrased variants via temperature-controlled decoding from LLMs, this paper measures the shifts in log-likelihood, entropy, and token frequency variance between original texts and paraphrased versions. These “shifts” features are then fed into an XGBoost model to yield interpretable decisions about authorship. Extensive experiments on PubMed, WritingPrompts, and XSum demonstrate that GramGuard matches or exceeds state-of-the-art detectors in performance and robustness.

**Strengths:**

1.	The idea of using pre-AI N-gram models trained solely on human-written corpora as a scoring model for AI text detection is interesting.
2.	The adopted XGBoost model helps improve the interpretability of the overall approach.
3.	The method achieves state-of-the-art detection accuracy across three datasets and maintains robustness under paraphrastic attacks.

**Weaknesses:**

1.	The proposed framework is not as lightweight as claimed in the paper since it involves large language models like GPT4 or Genimi for paraphrasing input samples.
2.	The paper only uses responses from GPT-4_1-mini as AI responses and paraphrases with GPT3.4, GPT 4 model, and Gemini model. Note that GPT-4_1-mini, GPT3.4 and GPT 4 are from the same model family. Experiments with synthesized AI responses from one more model will help strengthen the paper to have a more robust and diverse evaluation

**Questions:**

1.	The citation format is not correct. Using \citep{} and \citet{} appropriately.
2.	In Corollary 1.1, it says “Under perturbation, the process of rephrasing machine text tends to sample the tokens with lower probabilities compared with their original sample”. Is it true? And why?
3.	What is L in equation (6)?

---

> ### Author Response · Authors · 2025-11-20
> **Response to Reviewer A6yE**
>
> We thank the reviewer for the positive assessment of our work and for the constructive questions. We address each weakness and question in turn.
>
> **W1 – “Lightweight” framework vs. use of large LLMs for paraphrasing**
>
> We agree that our wording about “lightweight” could be clearer. GramGuard has two phases:
>
> 1. **Online paraphrase generation**, where we use large LLM-GPT-4 to create variants of each input; and
> 2. **Offline detection**, where only the pre-LLM n-gram LMs and a small XGBoost classifier are used.
>
> At *deployment time*, the detector itself runs entirely on CPU with pre-LLM n-gram models and XGBoost, and does not require any large LLM. The paraphrases can be pre-computed once, the core method does not depend on a specific generator. In the revision we will:
>
> - provide a short runtime analysis for the n-gram + XGBoost stage; and
> - soften the “lightweight” claim to emphasize that **the deployed detector is lightweight**, while paraphrase generation can be amortized or handled by external services.
>
> **W2 – Model family diversity (only GPT-4.1-mini / GPT-3.4 / GPT-4 / Gemini in the submission)**
>
> We appreciate this suggestion and agree that going beyond a single vendor family is important. Since the submission, we have run additional experiments with open-source LLMs, following exactly the same human/AI/paraphrase pipeline and feature extraction:
>
> - **LLaMA-3 70B**: ROC-AUC = 0.9559 (PubMed), 0.9418 (Writing), 0.9221 (XSum).
> - **GPT-J 6B**: ROC-AUC = 0.9446 (PubMed), 0.9657 (Writing), 0.9947 (XSum).
> - **OPT-2.7B**: ROC-AUC = 0.9713 (PubMed), 0.9818 (Writing), 0.9947 (XSum).
>
> These results indicate that GramGuard maintains **high detection performance across heterogeneous model families** (OpenAI, Gemini, LLaMA-3, GPT-J, OPT) and is not tied to a single API or architecture. In the revised manuscript we will (i) add these numbers to the main results table and (ii) include a short “Cross-LLM Robustness” subsection summarizing them.
>
> **Q1 – Citation format**
>
> Thank you for pointing this out. As also noted by another reviewer, we will carefully proofread the LaTeX source, correct the use of `\citet{}` vs. `\citep{}`, and ensure that all citations follow the ICLR style consistently.
>
> **Q2 – Statement in Corollary 1.1 (“rephrasing machine text tends to sample tokens with lower probabilities”)**
>
> We appreciate the opportunity to clarify this. Our intention was not to claim a universal theorem, but to summarize an empirical trend we observe: when we paraphrase LLM-generated text, the resulting variants often shift probability mass toward less likely n-grams under the *human-only* n-gram LM, compared with the original human text. This is what drives the delta features in Figure 1.
>
> We agree that the current wording is too strong. In the revision we will:
>
> - restate this as a **more modest proposition with explicit assumptions** (e.g., fixed human n-gram LM, finite paraphrase budget);
> - move the formal statement and proof sketch to an appendix; and
> - keep the main text focused on the **empirical evidence** (figures/tables) showing the systematic shift in log-likelihood, entropy, and variance between human vs. LLM text under paraphrasing.
>
> **Q3 – Meaning of \(L\) in Equation (6)**
>
> In our current submission, \(L\) is introduced earlier (before Eq. (3)) as the **sentence length (number of tokens)**, but this definition is easy to miss by the time the reader reaches Eq. (6). In the revision we will repeat the definition when Eq. (6) is introduced and add a small notation table collecting all key symbols, including \(L\), so that the notation is unambiguous.
>
> **Summary**
>
> In summary, we will (1) clarify that GramGuard’s deployed detector is lightweight (CPU-friendly n-gram LMs + XGBoost) and that large LLMs are only used offline for paraphrase generation, (2) add and highlight new results on LLaMA-3 70B, GPT-J 6B, and OPT-2.7B to demonstrate cross-LLM robustness, (3) fix citation style and explicitly define all symbols such as \(L\), and (4) restate Corollary 1.1 more carefully, presenting it as an empirical pattern with clear assumptions rather than a universal claim. We believe these changes address the reviewer’s concerns while preserving the main contribution of the paper.

---

### Note · Authors · 2025-11-21

I have read and agree with the venue's withdrawal policy on behalf of myself and my co-authors.